# Stable Fluorescence of Eu^3+^ Complex Nanostructures Beneath a Protein Skin for Potential Biometric Recognition

**DOI:** 10.3390/nano11092462

**Published:** 2021-09-21

**Authors:** Yue Zhao, Ziyu Yao, Christopher D. Snow, Yanan Xu, Yao Wang, Dan Xiu, Laurence A. Belfiore, Jianguo Tang

**Affiliations:** 1Institute of Hybrid Materials, National Center of International Joint Research for Hybrid Materials Technology, National Base of International Sci. & Tech. Cooperation on Hybrid Materials, College of Materials Science and Engineering, Qingdao University, 308 Ningxia Road, Qingdao 266071, China; zy17854274018@163.com (Y.Z.); qltu77@163.com (Z.Y.); 15684728280@139.com (Y.X.); wangyaoqdu@126.com (Y.W.); 17806247607@163.com (D.X.); belfiore@colostate.edu.cn (L.A.B.); 2Department of Chemical and Biological Engineering, Colorado State University, Fort Collins, CO 80523, USA

**Keywords:** Eu complex, oleic acid, fingerprints, PiFM, biometric recognition

## Abstract

We designed and realized highly fluorescent nanostructures composed of Eu^3+^ complexes under a protein coating. The nanostructured material, confirmed by photo-induced force microscopy (PiFM), includes a bottom fluorescent layer and an upper protein layer. The bottom fluorescent layer includes Eu^3+^ that is coordinated by 1,10-phenanthroline (Phen) and oleic acid (O). The complete complexes (OEu^3+^Phen) formed higher-order structures with diameter 40–150 nm. Distinctive nanoscale striations reminiscent of fingerprints were observed with a high-resolution transmission electron microscope (HRTEM). Stable fluorescence was increased by the addition of Eu^3+^ coordinated by Phen and 2-thenoyltrifluoroacetone (TTA), and confirmed by fluorescence spectroscopy. A satisfactory result was the observation of red Eu^3+^ complex emission through a protein coating layer with a fluorescence microscope. Lanthanide nanostructures of these types might ultimately prove useful for biometric applications in the context of human and non-human tissues. The significant innovations of this work include: (1) the structural set-up of the fluorescence image embedded under protein “skin”; and (2) dual confirmations of nanotopography and unique nanofingerprints under PiFM and under TEM, respectively.

## 1. Introduction

The technology to implant chips under the skin for recognition, for example, radio frequency identification [1,2], has provided the recognition requirement through the under skin method, but this implantable chip is hard and does not match flexible and elastic skin. Therefore, it is urgent and significant to develop technologies for flexible and biocompatible chips with a clear identifiable signal, which would be the extension of biometric recognition, to identify signals under skin from human and non-human sources (for example, pets and rare animals). To realize this purpose, the available materials will be most important. As a general strategy for designing the materials that are embedded under skin and interact with biological tissues, nanomaterials have increasing potential to match the needs of this area [3].

Fluorescent materials based on lanthanide (Ln^3+^) complexes [4,5,6,7] exhibit sharp emission peaks and high fluorescence intensity compared with traditional fluorescent organic molecules. The advantages of distinctive emission spectra and photostability of Ln^3+^ fluorescence nanomaterials make them suitable for biological applications [7], with the background of the widely investigated luminescence properties of Ln^3+^ complexes [8,9]. The emission intensity of Ln^3+^ complexes can be increased by the complexation of small organic conjugate molecules [10,11] through the antenna effect, and the emission intensity can be adjusted by varying ligands [12]. Eu^3+^ complexes can be used as luminescent materials because they typically have high fluorescence efficiency and a long excited state lifetime (microsecond timescale) [13] upon excitation with external energy [14]. Many small molecules have been demonstrated to serve as ligands of Eu^3+^ complexes, resulting in enhanced fluorescence intensities [15,16,17,18,19]. 1,10-Phenanthroline (Phen) is a favored ligand because Phen can form a stable bidentate interaction with Eu^3+^ with high fluorescence intensity due to the energy level match between the ligand triplet energy level and the first excited state level of europium ions [20,21,22]. 2-Thenoyltrifluoroacetone (TTA) and Phen have been successfully used to synthesize Eu^3+^ complexes with luminescence that is suitable for use in a solar concentrator context [23]. These features have attracted enormous attention [24,25]. In our group, we developed Ln^3+^-induced polymer aggregates [26], which work very well to securely embed lanthanide ions (Ln^3+^) within nanostructures. It is possible to use controllable nanostructures in the biosciences [27]. As an application example, europium complexes have been widely used in the field of bioimaging due to their intense red emission [28,29].

On the other hand, non-conjugated molecules (such as carboxylic acids, pyridine compounds, etc.) with coordinating groups (such as carboxyl or hydroxyl groups) as assisted ligands can also coordinate with Eu^3+^ ions to stabilize the structures of organic ligand–Eu^3+^ complexes [30].

However, we think that the single emitting property of materials provides a fluorescence signal that will have weak feature recognition ability, while combining multiple features improves the accuracy of biometric recognition. Therefore, we propose to develop materials that combine both the distinctive luminescence properties of Eu^3+^ complexes and the identifiable morphology of higher-order nanoassemblies thereof.

As oleic acid can coordinate with Eu^3+^ ions to stabilize the structures, it is easy to obtain. Therefore, in this work, we replaced the diblock copolymer mentioned above with a small amphiphilic organic molecule, oleic acid (O). The resulting assemblies included striated “fingerprint-like” nanostructures when observed via HRTEM. The constituent Eu^3+^ complexes, denoted as OEu^3+^Phen, were prepared via a facile method, using oleic acid (O) and Phen as ligands. The obtained Eu^3+^ complex was confirmed through FTIR spectra, PiFM, etc. The Eu^3+^ complexes showed good dispersion, great stability, and ligand ratio-dependent fluorescence intensity. Hypothetically, the distinctive OEu^3+^Phen striations could provide another form of material marking that is difficult to emulate. However, the only current method to observe striations is to use staining and HRTEM. Therefore, more convenient biometric recognition of these materials will rely on distinctive luminescence properties and topography, properties that can be maintained when the labelling material is covered by thin layers of biological molecules and tissues.

To assess recognition through intervening biological materials, we started by spin coating the OEu^3+^Phen nanostructures on a glass slide, amplifying luminescence by spin coating Eu^3+^(TTA)_3_phen, and then sequentially added 1–5 layers of protein (lysozyme was used as a protein material, via spin coating). We illuminated the resulting sample with 365 nm illumination on an inverted microscope. Bright red emission and stable fluorescence intensity from the OEu^3+^Phen layer could be observed from above via fluorescence microscopy through the intervening protein “skin” layer. We envision that devices could be made with these materials and offer multiple levels of authentication. First, luminescence provides an easy method for locating this class of labels at a distance. Unlike fluorescently labeled protein of any kind, these labels are not susceptible to proteolytic degradation. Unlike organic dyes, these labels are not susceptible to bleaching. Unlike most quantum dots, the luminescence lifetime of the lanthanide complexes is quite long (milliseconds). Second, labels that are located have nanotopography that can be inspected in situ (underneath layers of protein) using photo-induced force microscopy. Finally, if further authentication is necessary, labels could be harvested and stained with phosphotungstic acid, revealing distinctive striations that are absent on comparable materials with europium-based luminescence.

In summary, the OEu^3+^Phen complexes under a protein “skin” may ultimately enable biometric marking and recognition applications since they can provide multi-mode labels by combining characteristic fluorescence with unusual nanostructure features.

## 2. Materials and Methods

### 2.1. Materials

The Eu^3+^ source was from europium oxide (Eu_2_O_3_, purity 99.99%), which was purchased from Sinopharm Chemical Reagent Co. Ltd, Shanghai, China Eu_2_O_3_ and dissolved in dilute hydrochloric acid. The solution was transferred to an open heating glass dish to evaporate. The resulting powder, EuCl_3_(H_2_O)_6_, was dissolved in N, N-Dimethylformamide (DMF) and diluted to form a 0.02 mol L^−1^ solution for subsequent experiments.

### 2.2. Experimental Section

Synthesis of OEu^3+^Phen complex: First, 0.0198 g Phen was dissolved in 10 mL DMF in a 50 mL round bottom flask. The solution was heated in an oil bath to a constant temperature of 60 ℃. Second, a 15 mL solution of 0.05 mol L^−1^ oleic acid was added dropwise to the above solution with stable stirring. Third, 5 mL of 0.02 mol L^−1^ EuCl_3_ solution was added dropwise to the flask. The addition rate of EuCl_3_(H_2_O)_6_ was the same as that of oleic acid. Finally, the mixture was heated in an oil bath for 6 h. Finally, OEu^3+^Phen complexes were obtained.

Preparation of the OEu^3+^Phen fluorescence layer and protein skin layer: Five drops (about 0.15 mL) of OEu^3+^Phen complexes were dropped on the glass substrate and the glass substrate was spun for 30 s at 600 rpm. The process was repeated 3 times to coat the complexes evenly on the glass. Then, five drops of Eu(TTA)_3_Phen complexes were coated at the speed of 600 rpm for 30 s, to supplement the fluorescent intensity. The synthesis method of Eu(TTA)_3_Phen complexes was as described in a recent report [31].

Preparation of protein skin layer: 0.3 g protein (hen egg white lysozyme from Macklin) was dissolved in 10 mL distilled water. The concentration of protein solution was therefore 30 g/L. Five drops of protein solution (~0.15 mL) were deposited on the Eu^3+^ complex layers prior to spreading at 800 rpm for 20 s. Combined, the bottom lanthanide layer (including OEu^3+^Phen coating and Eu(TTA)_3_Phen complex coating) and top protein layers were prepared.

### 2.3. Characterization

The striated structure of OEu^3+^Phen aggregates was observed by transmission electron microscopy (TEM, JEM-1200EX, JEOL, Tokyo, Japan, when the Eu^3+^ complexes were stained by phosphotungstic acid [32] with volume ratio 1:1. The chemical bond composition and nanostructure of Eu^3+^ complexes under the protein were observed by photo-induced force microscopy (PiFM, Vista-IR, ST Instruments, Groot-Ammers, The Netherlands). The EDS mapping and EDS spectrum were obtained using a high-resolution transmission electron microscope (HRTEM, 2100-F, JEOL, Tokyo, Japan). The chemical bond composition and functional groups were characterized by Fourier transform infrared spectroscopy (FTIR, MAGNA-IR 550, Nicolet Instrument Corporation, Fitchburg, MA, USA) at room temperature. The UV–Vis absorption spectra of OEu^3+^Phen complexes was examined by UV–visible spectroscopy (UV–Vis, Lambda 750, PerkinElmer, Waltham, MA, USA). Photoluminescent (PL) spectra were obtained with a Cary Eclipse fluorescence spectrophotometer (Varian, San Francisco, CA, USA) at room temperature. The excitation slit width was 10 nm and the emission slit width was 10 nm. Excellent luminescence properties were confirmed by fluorescence spectroscopy (XSP-63XD, Shanghai Optical Instrument, Shanghai, China).

## 3. Results

### 3.1. Fingerprint-Like Nanostructures Characterized by TEM, HRTEM, and EDS Mapping

In previous work with diblock copolymer ligands, Eu^3+^ complexes adopted stable solid nanosphere structures as observed using TEM [32]. Here, we observed similar solid nanosphere structures for a low oleic acid concentration. Specifically, in Figure 1a, we show nanosphere structures with diameters around 200 nm. However, the aggregate structure changed from solid nanospheres to fingerprint-like nanostructures when the concentration of oleic acid increased from 0.02 mol L^−1^ to 0.05 mol L^−1^, keeping the other experimental conditions unchanged (Figure 1a–e). Individual nanofingerprint assemblies are about 70 to 140 nm in Figure 1b,c. The fine white and black lines that comprise the fingerprint-like nanostructures were about 2–3 nm in thickness. When the oleic acid concentration used during synthesis increased to 0.06 or 0.08 mol L^−1^, the fingerprint-like nanostructure could still be observed very clearly. The respective diameters of the two fingerprint images (Figure 1d,e) were about 120 nm and 160 nm, slightly larger than those in Figure 1b,c. Energy-dispersive X-ray spectroscopy (EDS) elemental mapping was utilized to observe the good dispersion of each element found in the Eu^3+^ complexes. Figure 1f indicates that Eu^3+^ (red dots), nitrogen (green dots), and phosphorus (blue dots) (from phosphotungstic acid) were well distributed in the fingerprint-like nanostructure (Figure 1f). The isolated Eu^3+^ (red dots) channel (Figure 1g) further shows that Eu^3+^ elements were distributed evenly in the corresponding nanosphere area. Given the sensitivity of the morphology to the oleic acid concentration, we hypothesize that the formation of nanofingerprints is directly related to the coordination interaction between the ligands and the Eu^3+^ ions. Figure 1h shows the EDS spectrum of each element in the OEu^3+^Phen complexes. Element peaks for C, O, Cl, and Eu^3+^ could be clearly observed in the sample. The Eu^3+^ content was very low due to the higher content of other ligands. Therefore, the Eu^3+^ complexes with fingerprint-like nanostructure can be obtained by controlling the concentration of the oleic acid, i.e., striated aggregate structures were obtained for oleic acid concentrations from 0.05 to 0.08 mol L^−1^. At high oleic acid concentrations, 0.1 mol L^−1^, no striations were observed.

### 3.2. UV–Visible Absorption Spectra Analysis

Figure 2 shows the UV spectra for OEu^3+^Phen, oleic acid, and Phen. Two strong Phen absorption peaks at 291 nm and 324 nm were observed clearly, ascribed to the π→π * transition and *n*→π * transition [33]. The ligand oleic acid has only one peak at 286 nm, which is ascribed to a *n*–π transition [34]. The Phen absorption peak red-shifts about 2 nm after complexing with Eu^3+^, indicating the coordination between Eu^3+^ ions with N atoms in Phen [35]. The UV absorption intensity of oleic acid is very low, but the addition of carboxyl groups and hydrophobic tails will change the spatial structure of the complexes, potentially improving the stability of the complexes [34].

### 3.3. Fluorescence Properties of OEu^3+^Phen Complexes

The excitation and emission spectra of isolated OEu^3+^Phen complexes are shown in Figure 3a. The broad excitation band around 329 nm is attributed to electronic transitions from the ground state level (π) to the excited level (π *) of the organic Phen ligand [36]. The ligand absorbed the energy of the excitation light in a π→π * transition. The excited energy was transferred to the europium ion, resulting in the fluorescence emission peak near 614 nm [37,38].

The OEu^3+^Phen emission spectra were obtained with excitation at 329 nm, shown in Figure 3a The strongest red fluorescence emission peak at 616 nm was attributed to the ^5^D_0_→^7^F_2_ electron transition [39]. Due to the lack of an antenna effect for the oleic acid ligands compared to the Phen, larger oleic acid concentrations led to weaker fluorescence intensity via an apparent binding competition with Phen. Combined, the UV and fluorescence spectra confirm formation of the expected OEu^3+^Phen complexes.

### 3.4. Observing Luminescent Nanostructures under a Protein Skin Layer by PiFM

Fundamental developments in nanotechnology can also promote the progress of applied biological fields. The OEu^3+^Phen complexes meet the requirements of the above nanotechnology with unique fingerprint-like nanostructures and distinctive fluorescence properties. We decided to evaluate the feasibility of using OEu^3+^Phen in a biorecognition context. Specifically, we tested whether OEu^3+^Phen could be observed through a thin protein layer (hen egg white lysozyme), used to mimic biological tissue. Lysozyme is a common protein and it has general characteristics, so we chose it as a protein material.

The FTIR spectra provide a detailed account of the chemical bond composition and functional groups present in the OEu^3+^Phen complexes, oleic acid, and protein, as shown in Figure 4a. In the protein sample, we attribute the 1648 cm^−1^ peak to amide I (primary amide N-H deformation vibration), and the 1526 cm^−1^ to amide II (secondary amide N-H deformation vibration). The 1710 cm^−1^ peak, observed as the blue line in Figure 4a, is the absorption peak corresponding to the C=O stretching vibration of the oleic acid carboxylate group [40]. However, the 1710 cm^−1^ oleic acid peak disappeared in the OEu^3+^Phen complexes while the strong peak at 1663 cm^−1^ appeared. Referencing similar publications, we infer that this shift was due to the coordination interaction between the carboxyl group ligand and Eu^3+^ [40,41,42]. For example, the investigations of L. A. Reinhardt found that the carboxylic stretch absorption peaks in oleic acid and trifluoroacetic acid have larger shifts, from 1710 cm^−1^ to 1673 cm^−1^, when the coordination to Eu^3+^ occurred [41]. Similarly, R. A. Harris indicated that the oleic acid absorption peak at 1710 cm^−1^ disappeared when oleic acid was chemisorbed onto Fe_3_O_4_ nanoparticles [42]. The above examples show that the FTIR carbonyl peaks can shift greatly in different reactions. Additionally, it can be seen that CH_3_ deformation vibrations occurred at 1381 cm^−1^ in the complexes [43]. We attribute the low-frequency peak (655 cm^−1^) in the complexes to Eu^3+^-O stretching vibrational modes [21].

Photo-induced force microscopy (PiFM) is a new nanoscale spectroscopy technology, which can not only show the morphological image but also the chemical composition distribution image for the same nanoarea of the same sample [44]. PiFM images provide a nanoscale spatial map of the individual chemical components in the sample [45].

In this work, PiFM was used to discern the morphology of a complex hybrid material that combined OEu^3+^Phen complexes with protein. The spectral responses for amide II of protein (1540 cm^−1^, Figure 4b) and carboxyl of OEu^3+^Phen complexes (1663 cm^−1^, Figure 4c) were marked with green correspondingly in each image, which also illustrate the nanotopography of the protein and OEu^3+^Phen complexes (red corresponds to low signal). Notably, the depth variations of this nanotopography (Figure 4d) are about 20 nm, which is much smaller than the height of the protein layers (~1500 nm, see below). We therefore propose that the visible nanospherical aggregates at 1663 cm^−1^ are visible despite being covered by the protein layer. The nanospherical aggregates have similar size and shape to the isolated aggregates seen via TEM (Figure 1b). Despite the presence of the protein layer, the size and shape of the OEu^3+^Phen aggregates could be observed from the PiFM images by tracking the oleic acid carboxylate IR signature.

The sample was measured by PiFM at three different points, as shown with yellow labels in Figure 4d. Corresponding spectra are shown in Figure 4e. The PiFM spectra for location #1 had characteristic peaks at 1690 cm^−1^, 1663 cm^−1^, 1670 cm^−1^, and 1540 cm^−1^. We attributed the distinctive 1690 cm^−1^ feature of the spectrum to the coordination bond formed between Eu^3+^ and carbonyl groups in TTA [46,47]. We attribute the absorption peaks at 1670 cm^−1^ (amide I) and 1540 cm^−1^ (amide II) to characteristic absorption features for the protein. As noted above, we can attribute the 1663 cm^−1^ peak to the carboxylate moiety of oleic acid [35]. Using an XSP-63XD fluorescence microscope with ~365 nm illumination, the sample had strong red fluorescence, demonstrating that Eu^3+^ complexes (OEu^3+^Phen and Eu(TTA)_3_Phen complexes) could be detected through the protein coating layer as shown in Figure 4f. The addition of the Eu(TTA)_3_Phen layer was only intended to improve the fluorescence intensity of the sample. As can be seen from Figure 4c, the addition of Eu^3+^(TTA)_3_Phen layer did not prevent the observation of nanostructures that are consistent with the size of with size of OEu^3+^Phen complexes observed via TEM. Therefore, the Eu(TTA)^3^Phen complex will aid rather than hinder biorecognition.

### 3.5. The Effect of Different Thicknesses of Protein Layers

In order to explore the effect of protein layer thickness on luminescence properties, one to five layers of proteins were prepared by spin coating. The sample altitude of just the OEu^3+^Phen complex layer was measured by Atomic Force Microscope (AFM) before the addition of the protein layers, as shown in Figure 5a. The brighter spots are our fingerprint nanoparticles. We took some of these points and analyzed it, as shown in Figure 5b. We found that it was about 20 nm in height. The thicknesses of the protein layers prepared by spin coating were measured by surface profiler (Dektak 150, Veeco, New York, NY, USA), as shown in Table 1. We calculated the thickness of the protein layer from the difference between total sample thickness and the thickness of the OEu^3+^Phen complex layer (recorded in Table 1). We calculated that each layer of protein was about 300 nm thick. The fluorescence masking effect of the protein layers was quantified with a fluorescence spectrophotometer (Figure 5c). We found that the fluorescence emission decreased very gradually as the protein layer thickness was increased. The fluorescence lifetime and absolute quantum yield were measured, as shown in Table 2. The fluorescence lifetime was fitted by the following formula: A + B_1_e^(−t/τ1)^ + B_2_e^(−t/τ2)^, as shown in Figure 5d. We found that the protein skin induced no significant change in the luminescence lifetime of the underlying lanthanide layer, which remained around 0.8 ms. Absolute quantum yield was also maintained at around 70%. Up to a depth of 1.5 microns, the thickness of the spin coated protein layer has negligible effect on the luminescence performance.

### 3.6. Prognosis for Biometric Recognition Applications

Lanthanide complexes offer a unique mixture of benefits and drawbacks with respect to detection through biological tissues. Foremost, the red emission wavelength of Eu^3+^ complexes is advantageous due to greater penetration of tissues. However, UV excitation is a distinct disadvantage unless the labels are close to the surface, since biological tissues will absorb UV light and will exhibit autofluorescence. However, compared to typical fluorophores, lanthanide luminescence is very long-lived, allowing for the elimination of background emissions via time gating. In sum, europium complexes offer potential utility to provide fluorescent labels through intervening protein layers, which may be used for potential biometric recognition applications. A simple multiple biometric recognition concept is outlined in Figure 6. To prove the feasibility of this concept, glass was first coated with OEu^3+^Phen and Eu(TTA)_3_Phen complexes, followed with a protein layer intended to crudely mimic the epidermis (Figure 6). The large Stokes shift, the distinctive emission spectrum, the distinctive size and shape of the lanthanide complex aggregates, and the unique chemical bonds of the constituent molecules can be recognized by combining fluorescence microscopy and PiFM characterization through layers of protein. The OEu^3+^Phen and Eu^3+^(TTA)^3^Phen complexes showed red fluorescence, which could be observed with a transmission illumination (365 nm) fluorescence microscope through protein layers. The oleic acid carboxylate groups of the OEu^3+^Phen complexes were identified by PiFM despite an intervening protein layer. Finally, for an ultimate level of authentication, we propose that the distinctive nanoscale striations present at the atomic scale in the lanthanide marks could be observed by excising the marks from the simulated biological tissue, staining, and subjecting the samples to HRTEM. Together, these techniques illustrate the potential for highly recognizable markers. Finally, we show that marker materials could be rapidly identified at a distance using the large Stokes shift luminescence and could be further confirmed in detail via the characteristic size and shape of the lanthanide complex aggregates. Future work should further probe the biometric recognition potential of these materials via combining them with increasingly realistic models of biological tissues (e.g., the epidermis).

## 4. Conclusions

In summary, we successfully synthesized OEu^3+^Phen complexes that adopt a distinctive nanofingerprint structure as assessed via TEM and yield stable luminescence output based on the Ln^3+^-induced polymer aggregation principle that we established previously in our group. When the concentration of oleic acid was 0.05 mol/L, the complex had a distinctive nanofingerprint structure and stable fluorescence intensity. The formation of nanofingerprint structures was sensitive to the concentration of oleic acid. The properties of the complexes were investigated by TEM, spectroscopy, and PiFM. We demonstrated that red luminescence and morphology features of OEu^3+^Phen complexes could be detected through multiple protein layers with a cumulative thickness of ~1.5 microns. With further development, OEu^3+^Phen fluorescent complexes may have potential uses for discrete labeling and authentication. New luminescent materials will help meet emerging needs in the biosensing and bioimaging fields.

## Figures and Tables

**Figure 1 nanomaterials-11-02462-f001:**
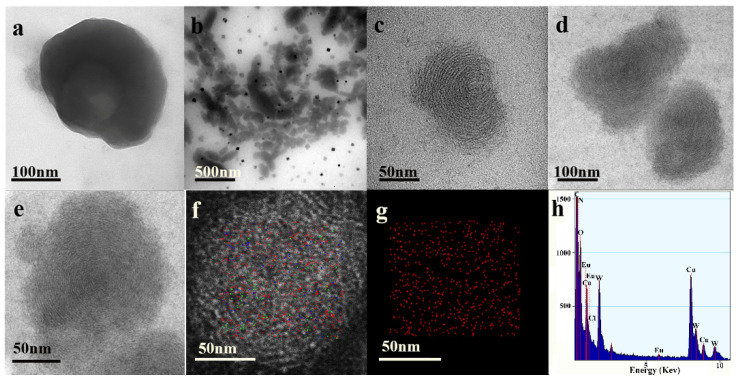
TEM and HRTEM images (**a**–**e**) of the OEu^3+^Phen complex under different magnifications in the different concentrations of oleic acid: 0.02 mol L^−1^ (**a**), 0.05 mol L^−1^ (**b**,**c**), 0.06 mol L^−1^ (**d**), and 0.08 mol L^−1^ (**e**), respectively. (**f**,**g**) EDS mapping images of the fingerprint-like nanostructures for 0.08 mol L^−1^ oleic acid, and (f) shows all the elements of Eu (red), N (green), P (blue), while (**g**) shows only Eu^3+^ element. (**h**) The EDS spectrum of the OEu^3+^Phen complexes for 0.05 mol L^−1^ oleic acid.

**Figure 2 nanomaterials-11-02462-f002:**
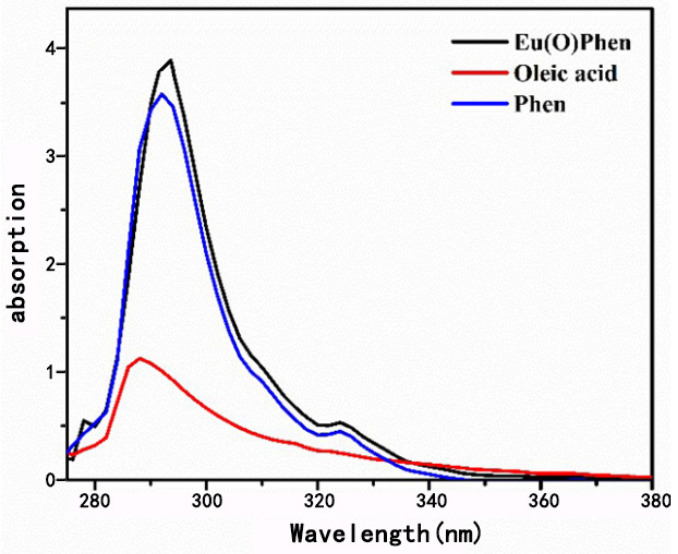
UV–Vis absorbance spectra of OEu^3+^Phen complex, oleic acid, and Phen.

**Figure 3 nanomaterials-11-02462-f003:**
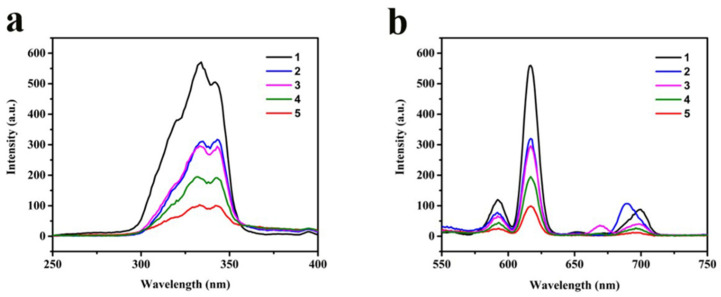
(**a**) Excitation spectra and (**b**) emission spectra of OEu^3+^Phen complexes in DMF solutions. The concentrations of oleic acid from spectra #1 to #5 were 0.02, 0.04, 0.05, 0.06, and 0.08 mol L^−1^, respectively.

**Figure 4 nanomaterials-11-02462-f004:**
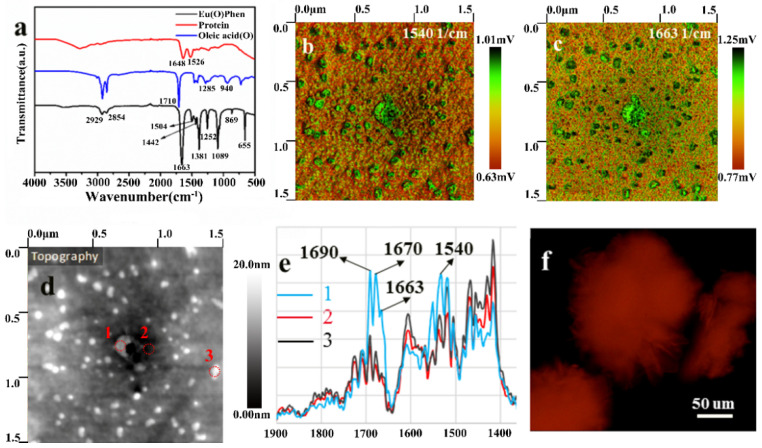
The concentration of oleic acid used during nanolabel synthesis was 0.05 mol L^−1^ in FTIR spectra and PiFM images. (**a**) FTIR spectra of protein, oleic acid, and OEu^3+^Phen complexes; PiFM images taken with laser tuned to (**b**) 1540 cm^−1^ for protein and (**c**) 1663 cm^−1^ for OEu^3+^Phen complexes in the sample; (**d**) the detection points of the sample, overlaid on a topographical map; (**e**) PiFM infrared spectra for the detection points; (**f**) the fluorescence microscopy image of the sample.

**Figure 5 nanomaterials-11-02462-f005:**
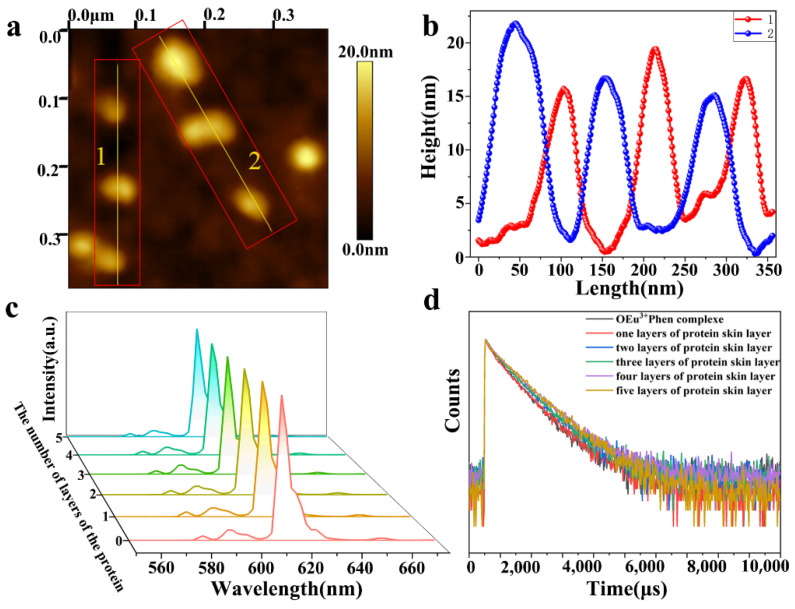
(**a**) AFM image of OEu^3+^Phen complex layer; (**b**) the local height in Figure a; (**c**) fluorescence intensity of 0 to 5 layers of protein; (**d**) the fluorescence lifetime curve of protein from 0 to 5 layers.

**Figure 6 nanomaterials-11-02462-f006:**
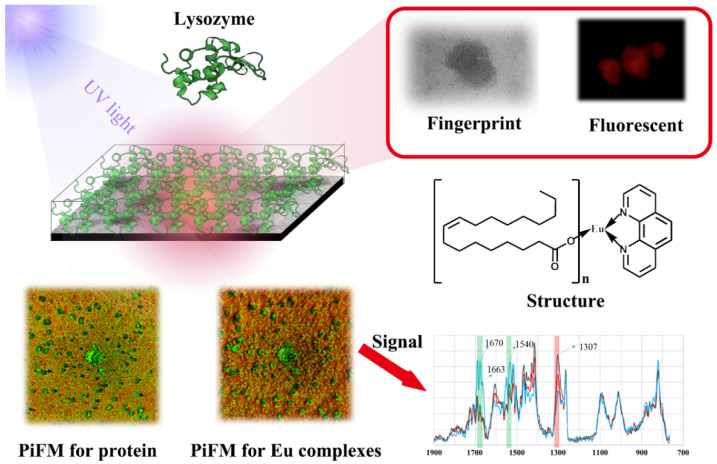
Principle diagram of Eu^3+^ complexes implanted under protein.

**Table 1 nanomaterials-11-02462-t001:** Height of 1–5 layers of protein skin layer.

	Height/nm	Height of Protein/nm
OEu^3+^Phen complex layer	20	
One layer of protein skin layer	361	341
Two layers of protein skin layer	665	645
Three layers of protein skin layer	946	926
Four layers of protein skin layer	1256	1236
Five layers of protein skin layer	1584	1564

**Table 2 nanomaterials-11-02462-t002:** Fluorescence lifetime parameters and absolute quantum yield.

τ1/ms	τ2/ms	B1	B2	Lifetime/ms	Quantum Yield
0.6708	0.8254	20.26%	79.74%	0.79408	64.17%
0.7444	0.8097	24.59%	75.41%	0.79365	66.32%
0.6654	0.8393	29%	71%	0.78886	66.96%
0.6924	0.8115	19.26%	80.74%	0.78857	67.09%
0.5617	0.8596	24.01%	75.99%	0.78808	67.30%
0.6283	0.8282	20%	80%	0.78822	68.01%

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
