# Peer review of "Stable Fluorescence of Eu3+ Complex Nanostructures Beneath a Protein Skin for Potential Biometric Recognition"

_nanomaterials, 2021, doi:10.3390/nano11092462_

Round 1

Reviewer 1 Report

The authors of present work synthesized fluorescent nanoparticles of Eu3+ complexes with 1,10-phenanthroline and oleate as ligands. The nanostructures were characterized with various methods. Fluorescence parameters of this particles were shown to be relatively stable after coating with protein layers. Taking into account emission of lantanide containing nanoparticles in red and NIR region of spectra, such methodology could be useful for development of noninvasive imagine and diagnostics procedures. The manuscript is clearly and well written and is recommended for publication. However, some minor corrections needed:

1) What was the reason for choosing of oleic acid as ligand? Author should comment their choice in the text.

2) Authors should give additional references on reviews or books in the introduction. For example, in addition to references [1-3].

3) The curve on Fig.2. seems to be broken near the maximum. Probably, the it was depicted with 5 nm step. However, in this case the differences in absorption maxima between Phen and OEuPhen insignificant. Authors should correct the Fig. 2.

4) References [1] and [2] are not correctly formatted (lines 349-356)

5) Table 2 is not correct. There no lifetime and QY.

Reviewer 2 Report

Stable Fluorescence of Eu3+ Complex Nanostructures Beneath a Protein Skin for Potential Biometric Recognition

Yue Zhao 1, Ziyu Yao 1, Christopher D. Snow 2,*, Yanan Xu 1, Yao Wang 1, Dan Xiu 1, Laurence A. Belfiore 1,2, and Jianguo Tang 1,*

This paper describes the use of Europium complex nanostructures as potential biometric applications in the context of human or non-human tissues. The authors show the unique features of these nanostructures with the use of multiple techniques.

I am not fully convinced of all the results that are provided to make a statement that the nanostructures can be used as biometric marker. In this paper the authors use multiple protein layers as mimic for skin, at least that is how I interpret it. The thickness of 1 protein layer is about 300 nm and, in their experiments, a maximum thickness of 1.5 um is achieved. Skin on the other hand can vary from 50 um to 1.5 mm. This is way beyond the thicknesses of the protein layer and to claim that the developed nanostructures have potential in biometric applications is a stretch.

I appreciate that the authors have used different technologies to characterize the nanostructures. However, I do have some comments/questions/remarks which will be listed below.

Textual  points;

The text of the paper is sometimes difficult to grasp. for example the first paragraph of the introduction is unreadable “Information monitoring under skin is highly recommended for recognition in order to identify individual characteristics. However, the currently used sensing method requires collection of real samples from true site under skin, which is painful and uncomfortable surgery. Luminescence behind thin protein skin can provide information that has been existed under the skin, when the luminescence can be detected outside the skin. This is a kind of sensing way.” I have no idea what the authors want to convey here. Please rewrite and have a thorough read of the current manuscript.

Line 51, “and available morphologies” sounds little strange. What do the authors mean with this?

Line 75 “confirmed through measurements”. I would mention a few techniques that were done.

Line 85 is mentioned “1-5 layers of protein”. Here I would mention which protein you have used. The authors are specific in describing the complex nanostructure.

Line 86 “We illuminated the resulting sample from below with 365 nm illumination”. Strange wording, mention first that you use an inverted microscope, which is clear that you illuminate from below.

Line 91 The authors mention “unlike fluorescent proteins”. The authors probably do not refer to fluorescent protein family (GFPs) because this protein is very stable and do not degrade and then this statement cannot be made. If the authors mean fluorescently labelled protein of any kind, that particular protein can of course be susceptible for proteolytic degradation.

Line 147. The authors use both the terms TEM and HRTEM. What is exact the difference between these systems? Is it another device?

Line 184 The authors mention “The ligand oleic acid has only one strong peak at 286 nm, which is ascribed to a n-π transition[31]. The Phen absorption peak red-shifts about 2 nm after complexing with Eu3+, indicating the coordination between Eu3+ ions with N atoms in Phen[32].” First, the term "strong peak" is little odd. A clear peak at 286 nm I would say. Furthermore, the absorption sepctrum shown in Figure 2 should be improved. On the Y-axis I would label it absorption and include the values on the Y-axis. Next, the authors mention a shift of 2 nm after complexation. Looking at the spectra I wonder if this conclusion can be drawn as the spectra appear to be recorded with a scan resolution of about 5 nm.

Line 193; Usually absorption spectra and excitation spectra are more or less the same but here this is here not the case. At what wavelength was the emission set to record the excitation spectra?

Line 215 The authors write “ The lysozyme is a typical representative of protein and it has the advantage of wide distribution”. This is a very strange sentence and could be extended/improved including a reference as it has the authors mention the diverse applications apparently lysozyme is involved in.

Line 272, the heading “3.5. The effect of different protein layers” is misleading, I would write multiple protein (lysozyme) layers. In line with this, have the authors used other protein to create a protein layer? If this is a generic application, I am curious to see what nanostructure is obtained with another protein.

Round 2

Reviewer 2 Report

The authors have modified the ms appropriately and can be published.